# A cryo-electron tomography study of ciliary rootlet organization

Chris van Hoorn*, Andrew P Carter*

MRC Laboratory of Molecular Biology, Cambridge, United Kingdom

## eLife assessment

The study offers a **compelling** molecular model for the organization of rootlets, a critical organelle that links cilia to the basal body, ensuring proper anchoring. While previous research has explored rootlet structure and organization, this study delivers an unprecedented level of resolution, **valuable** to the centrosome and cilia field. This research marks a significant step forward in our understanding of rootlets' molecular organization.

*For correspondence:
chrisvanhoorn@gmail.com (CH);
cartera@mrc-lmb.cam.ac.uk
(APC)

Competing interest: The authors declare that no competing interests exist.

**Abstract** Ciliary rootlets are striated bundles of filaments that connect the base of cilia to internal cellular structures. Rootlets are critical for the sensory and motile functions of cilia. However, the mechanisms underlying these functions remain unknown, in part due to a lack of structural information of rootlet organization. In this study, we obtain 3D reconstructions of membrane-associated and purified rootlets from mouse retina using cryo-electron tomography. We show that flexible protrusions on the rootlet surface, which emanate from the cross-striations, connect to intracellular membranes. In purified rootlets, the striations were classified into amorphous (A)-bands, associated with accumulations on the rootlet surface, and discrete (D)-bands corresponding to punctate lines of density that run through the rootlet. These striations connect a flexible network of longitudinal filaments. Subtomogram averaging suggests the filaments consist of two intertwined coiled coils. The rootlet's filamentous architecture, with frequent membrane-connecting cross-striations, lends itself well for anchoring large membranes in the cell.

## Introduction

Cilia are antenna-like cellular protrusions that function as sensory signaling hubs (primary cilia) (*Malicki and Johnson, 2017*) or motile propellers (motile cilia) (*Spassky and Meunier, 2017*). Cilia are connected to centrioles (referred to as basal bodies) which are in turn connected to striated cytoskeletal fibers called rootlets (*Engelmann, 1880*). Rootlets are classified into protist and metazoan types which differ in striated repeat distance, protein composition (*Andersen et al., 1991*), and functional properties (*Potter et al., 2017*; *Hayes et al., 2021*).

Rootlets are essential for the proper functioning of both primary and motile cilia (*Yang et al., 2002*; *Yang et al., 2005*). The first evidence for a role in primary cilia came from a partial knockout of the main constituent of rootlets, rootletin (CROCC), in mouse models. This resulted in late-onset blindness (*Yang et al., 2005*) and increased fragility of the cytoskeleton inside the primary cilium that links the inner and outer segments of photoreceptor cells (*Yang et al., 2005*; *Gilliam et al., 2012*). Subsequently, knockout of the rootletin homolog in *Drosophila* (Root) abolished neuronal primary cilia responses such as chemosensing, and touch sensitivity (*Chen et al., 2015*). Rootlets are also implicated in mechanosensory signal relay following primary cilia bending in fish (*Gilbert et al., 2021*). In the case of motile cilia, studies in *Xenopus* showed that disruption of a ciliary adhesion complex that tethers rootlets to cortical actin led to disorganized ciliary arrays (*Antoniades et al., 2014*; *Yasunaga*

*et al., 2022*). Taken together, rootlets appear to provide stability and correct anchorage for primary and motile cilia. Separately, rootlet-like structures are also found in non-ciliated cells where they tether centriole pairs together during the cell cycle (*Meraldi and Nigg, 2001*; *Vlijm et al., 2018*; *Yang et al., 2006*).

Rootlets are bundles of filaments with a total width of up to 300 nm (*Yang et al., 2002*), being at their widest where they contact centrioles from which they taper into the cell (*Spira and Milman, 1979*; *Vlijm et al., 2018*). They are characterized by regular cross-striations along their length as observed in resin-sectioning electron microscopy (EM) and immunofluorescence (*Olsson, 1962*; *Hagiwara et al., 2000*; *Vlijm et al., 2018*). The spacing of striations varies between organisms, with a reported 60 nm repeat in guinea pig (*Olsson, 1962*) and 80 nm in human photoreceptor cell rootlets (*Gilliam et al., 2012*). Variations of striation repeat length can also occur within single rootlets. For example, in human oviduct cilia, a 1- to 2-µm segment closest to the centrioles contains a 150- to 200-nm repeat that abruptly transitions to a 75- to 100-nm repeat for the rest of the structure (*Hagiwara et al., 2000*). The functional relevance of the striations remains unknown, but they may be involved in cellular interactions. For instance, mitochondrial cristae have been observed to align with striations when mitochondria contact rootlets (*Olsson, 1962*; *Hayes et al., 2021*). Other membrane connections are suggested from EM cross-sections of resin-embedded rootlets which reveal they are surrounded by membrane saccules (*Spira and Milman, 1979*).

The filaments which make up rootlets are assumed to be comprised of rootletin (*Yang et al., 2002*; *Surkont et al., 2015*). This coiled-coil protein is predicted to have a length of >200 nm and immunofluorescence shows its N- and C-termini align in repeating interdigitated bands along the rootlet (*Vlijm et al., 2018*). Rootletin contains four predicted coiled-coil domains (CC1–4) of which CC2, 3, and 4 were shown to homodimerize (*Yang et al., 2002*; *Ko et al., 2020*). There is currently disagreement as to whether CC1 is a coiled coil or a globular domain and whether it contributes directly to striation formation (*Yang et al., 2002*; *Akiyama et al., 2017*; *Ko et al., 2020*). Studies on non-ciliated cell rootlets suggest the other main components are CEP68 and CCDC102B which immunoprecipitate with rootletin (*Vlijm et al., 2018*; *Xia et al., 2018*). CEP68 is predicted to be mostly unstructured with an N-terminal three-helix spectrin-like fold. Immunofluorescence suggests it is found regularly spaced along the rootlet (*Vlijm et al., 2018*). CCDC102B contains three coiled-coil stretches, with a total length of approximately 42 nm, separated by disordered regions. Depletion of either CEP68 or CCDC102B results in thinner splayed rootlets, suggesting they are required for bundling the rootlet filaments (*Vlijm et al., 2018*; *Xia et al., 2018*). Additional components link rootlets to cellular structures. C-Nap1 (CEP250) is a long coiled-coil protein and paralog of rootletin (*Fry et al., 1998*; *Chen et al., 2015*) which connects rootlets to the base of centrioles (*Fang et al., 2014*; *Vlijm et al., 2018*; *Mahen, 2021*). At the other end, the outer nuclear membrane protein Nesprin1α links the rootlet to the nucleus (*Potter et al., 2017*).

Despite our knowledge of rootlet components, our 3D understanding of how rootlets are formed is limited. A key question is what the individual rootlet filaments look like and how they pack within the rootlet network. Moreover, it is not known what features of the ultrastructure give rise to regular striations. To address these questions, we used a partially purified sample and further purified native rootlets from mouse photoreceptor cells and assessed their organization with cryo-electron tomography (cryo-ET).

## Results

### Purification of mouse photoreceptor cell rootlets

The retinas of the eyes contain photoreceptor cells made up of inner and outer segments joined by a short primary cilium (*Figure 1A*). At the base of this cilium, connected to its basal bodies and running through the inner segment, is one of the longest observed rootlets (*Yang et al., 2002*). We isolated rootlets attached to cell outer segments according to *Gilliam et al., 2012*. In brief, retinas were dissected from mouse eyes and stripped from the associated retinal pigment epithelium. The outer segments, together with connecting cilia and rootlets, were dissociated from these by vortexing (*Figure 1B*). This sample was enriched using an Optiprep step gradient, applied to EM grids and plunge frozen for imaging by cryo-EM.

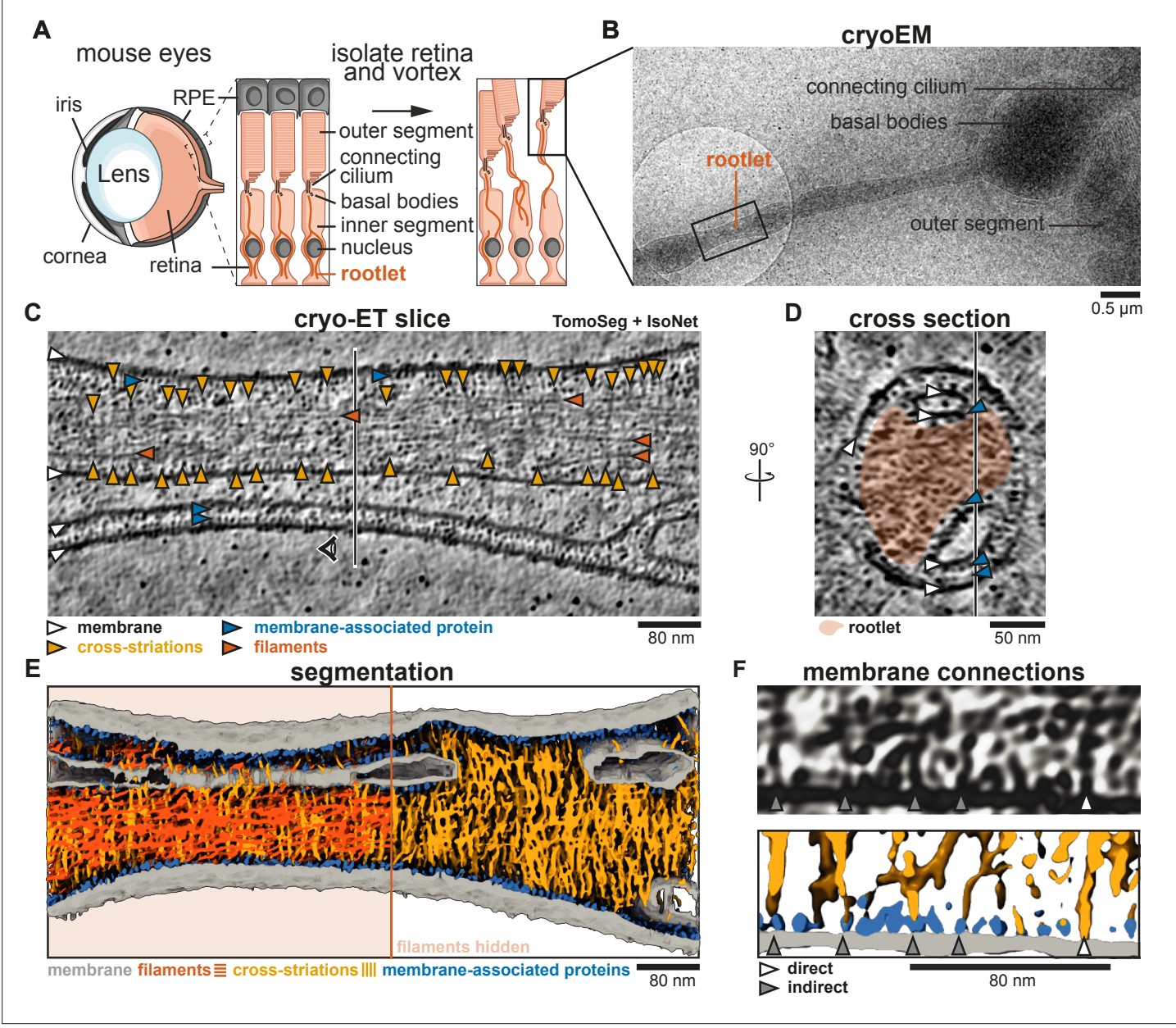

**Figure 1.** Cross-striations protrude from rootlets and connect to intracellular membranes. (**A**) Schematic of photoreceptor cell isolation from mouse eyes. Insets show the interface between photoreceptor cells and retinal pigment epithelium (RPE). (**B**) Low-magnification cryo-electron tomograph (cryo-EM) of the isolated sample. The black square depicts the location of acquisition of panel C. (**C**) Slice through a denoised and isotropically reconstructed cryo-electron tomogram. The vertical line indicates the position of the cross-section in panel D. (**D**) Cross-section of panel C, with the position of panel C indicated by a vertical line. The rootlet is highlighted in orange. (**E**) Tomogram segmentation using Eman2 TomoSeg (*Chen et al., 2017*) with filaments displayed in orange on the left half and hidden on the right. (**F**) Example of a tomogram slice with membrane connections, and their corresponding segmentation, showing direct membrane connections and connections via membrane-associated proteins with gray and white arrows, respectively.

The online version of this article includes the following figure supplement(s) for figure 1:

**Figure supplement 1.** Analysis of cross-striations in cellular tomograms.

To prevent sample deformation, the outer segment and basal body sections were typically embedded in thick ice, limiting their visibility. However, the protruding rootlets were in thinner ice allowing their imaging (*Figure 1B*). We collected cryo-electron tomograms to analyze their morphology. For improved contrast, we denoised the tomograms using Noise2Map (*Tegunov and Cramer, 2019*), followed by deconvolution and isotropic reconstruction using IsoNet (*Liu et al., 2021*).

## Cross-striations contact membranes

As previously described (*Gilliam et al., 2012*), rootlet striation repeat and morphology appear unaltered by the purification method. Moreover, they are proposed to maintain aspects of their native interactions with cellular membranes (*Gilliam et al., 2012*). To understand how the filaments and striations in rootlets connect to membranes, we turned to high-resolution cryo-ET.

In the two tomograms we analyzed in detail, we observed rootlets (*Figure 1C, D*, *Figure 1—figure supplement 1A*) flanked by membranes, vesicles, and stretches of the plasma membrane. Although the tomograms were noisy we could identify longitudinal filaments (orange arrowheads in *Figure 1C*, *Figure 1—figure supplement 1A*) and some cross-striations (*Figure 1C*, *Figure 1—figure supplement 1A*, yellow arrowheads). We realized that the cross-striations are faint in single slices and therefore performed a segmentation of the entire tomogram (*Figure 1E*) by training the TomoSeg convolutional neural network of eman2 (*Chen et al., 2017*). This revealed many perpendicular densities that weave between the longitudinal filaments, forming an intricate network.

All the striations partially or fully spanned the width of the rootlet and extended beyond the outermost longitudinal filaments. These rootlet-protruding striation densities frequently contacted the membrane (*Figure 1E*). Close examination suggested some make a direct contact, whereas others contact a subset of globular membrane-associated densities that are a striking feature of the tomograms. These densities are ~7 nm in diameter and cover almost every membrane surface. Where membranes come into proximity, the intervening space is filled with two layers of these membrane-associated proteins, one layer associated with each membrane (*Figure 1C*, *Figure 1—figure supplement 1A*, blue arrowheads). We trained a TomoSeg neural network to assign these densities and let this network compete with one that assigned striations. This resulted in a final segmentation with membrane-associated densities indicated in blue and striations in yellow (*Figure 1E, F*, *Figure 1—figure supplement 1D–F*). The proteinaceous protrusions that extended from the rootlets were flexible and did not induce a regular spacing in the membrane-associated proteins they contacted (*Figure 1F*, *Figure 1—figure supplement 1D–F*). Thus, densities that contribute to the cross-striations protrude from the rootlet and appear able to flexibly tether it to its surrounding membranes.

## Rootlets contain two types of cross-striations

Due to the ice-thickness and the presence of membranes, the tomograms had limited contrast. This made the repeat pattern of cross-striations observed in conventional, resin-embedded EM hard to see. To improve their visualization, we further purified rootlets based on a modified protocol from *Liu et al., 2007*. This involved detergent-solubilizing the membranes and sedimenting the rootlets through a sucrose cushion (*Figure 2A*). Cryo-EM of the resulting pellet showed clear rootlets with cross-striations and associated basal bodies (*Figure 2B*, *Figure 2—figure supplement 1A–C*). However, we noticed the rootlets tended to aggregate in clusters, regardless of buffer condition (*Figure 2—figure supplement 1A*). To overcome this, we vortexed the sample before grid preparation resulting in a sufficient number of single rootlets, separated from contaminating debris (*Figure 2B*).

We initially assessed the sample by negative stain EM which highlights features on the rootlet surface. A repeat pattern is immediately obvious, revealing features that have not been observed before. Each repeat consists of two types of bands: two discrete (D)-bands, which we named D1 and D2, and one wider, amorphous (A)-band (*Figure 2C*). The A-bands were frequently associated with tufts of material that extended away from the rootlet (*Figure 2C*).

We were also able to see these bands with cryo-ET. The striations in the purified rootlets appeared more ordered and clearer than in the cellular tomograms due to the improved contrast. In the cellular rootlets, we identified the bands in a tomogram projection (*Figure 1—figure supplement 1B*), with an average distance of 79.52 ± 0.26 nm between each repeat (*Figure 1—figure supplement 1C*). The repeat distance for the purified rootlets is 80.1 ± 0.03 nm based on a sine fit to A- and D-bands of 10 Fourier-filtered tomogram projections (*Figure 2D*, *Figure 2—figure supplement 1E–I*). This distance

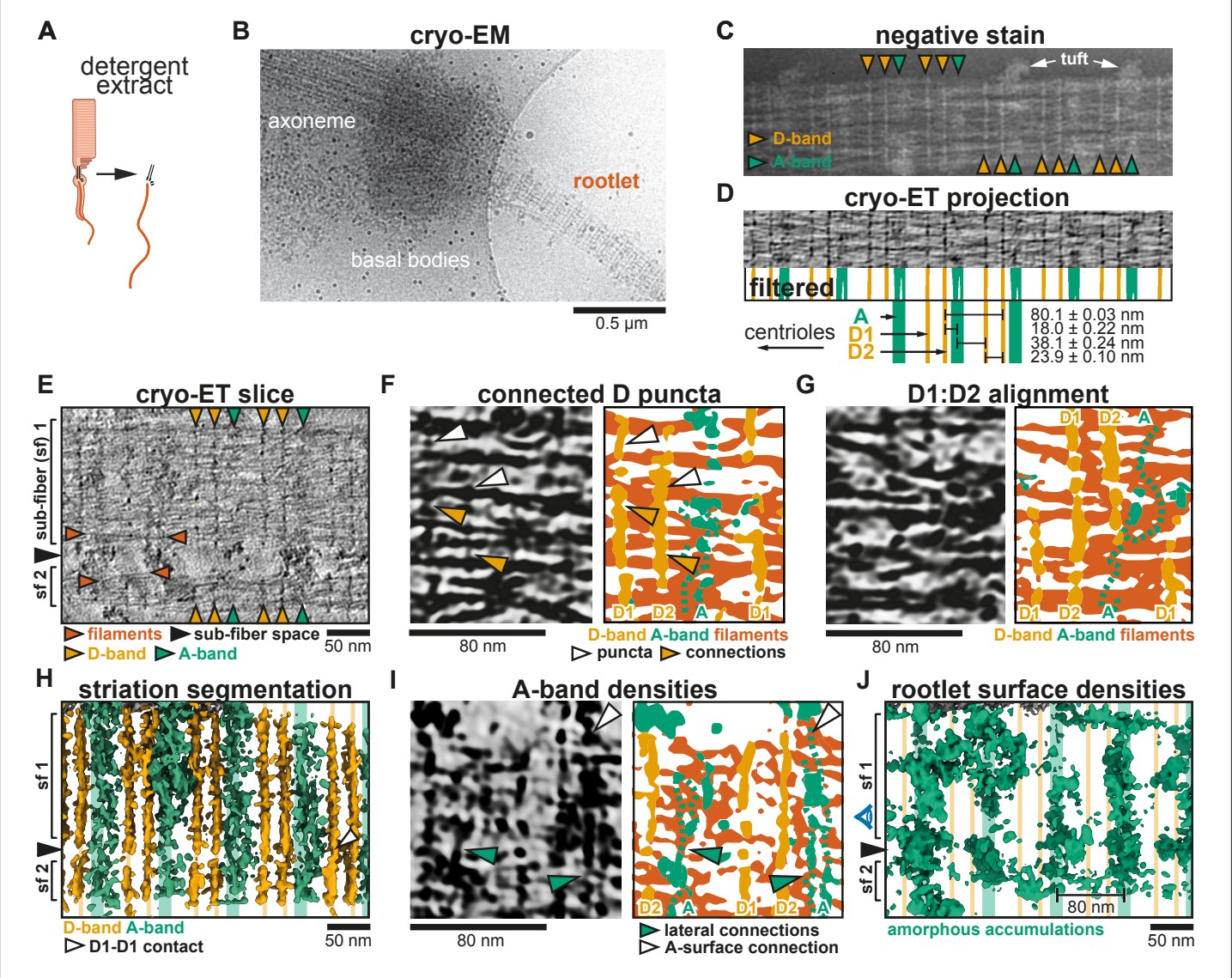

**Figure 2.** Cryo-electron tomography (cryo-ET) analysis of rootlet striations. (**A**) Schematic depiction of rootlet purification by membrane removal and gradient centrifugation (not shown). (**B**) Low-magnification cryoEM micrograph of a purified rootlet and associated ciliary cytoskeleton. (**C**) Negative stain EM of a purified rootlet, highlighting features visible on the rootlet surface. (**D**) CryoET projection image of a purified rootlet. The Fourier-filtered and thresholded striations are colored according to their appearance: D-bands in yellow and A-bands in green. Mean values of their spacing and the location of the centriole are indicated below, based on *Figure 2—figure supplement 1H–J*. (**E**) Central slice in a denoised and isotropically reconstructed electron tomogram showing two rootlet sub-fibers. (**F**) Example of fine features of D-bands in a cryo-ET slice and its segmentation (**G**) Example where D1 aligns with D2 of a neighboring sub-fiber. Larger view in *Figure 2—figure supplement 1A*. (**H**) Segmentation of the striations in the tomogram from panel D. D1–D1 contact of two sub-fibers is indicated by a white arrow. (**I**) Segmentation that shows amorphous features occur as two bands and connect to the rootlet surface densities. (**J**) Segmentation of amorphous material on the rootlet surface. The side view is shown in *Figure 2—figure supplement 2H* (**H, J**) The position of the A- and D-bands is shown by lines in the background. (**E, H, J**) Black arrows indicate the space between sub-fibers. (**F, G, I**) Fainter features not picked up by the automated segmentation were drawn with dotted lines.

The online version of this article includes the following figure supplement(s) for figure 2:

**Figure supplement 1.** Cross-striation analysis of purified rootlets.

**Figure supplement 2.** Segmentation of purified rootlets.

is consistent with measurements from conventional EM in mouse and human rootlets (*Gilliam et al., 2012*; *Vlijm et al., 2018*). The banding pattern is asymmetric within the 80.1 nm repeat. The D1- and D2-bands are 23.9 nm apart, followed by an 18.0-nm distance between D2- and the A-band. Between the A-band and D1 of the next repeat we measured a 38.1-nm distance (*Figure 2D*, *Figure 2—figure*

*supplement 1H, J*). We examined three rootlets in which either the centriole or rootlet tip was visible (*Figure 2—figure supplement 1B–D*). In all three cases, the D1-band is closest to the centriole and the A-band to the tip. This polar arrangement of striations can be used as a marker for rootlet orientation and supports the previously proposed polar assembly of the rootlet (*Bahe et al., 2005*; *Vlijm et al., 2018*).

## The D-bands are punctate and connect filaments

The D-bands are clearly visible in tomogram slices through the center of purified rootlets (*Figure 2E*, *Figure 2—figure supplement 2B*) revealing they span throughout the interior. We see them embedded within the array of longitudinal filaments that run along the rootlet's length. Both D1 and D2 are made up of punctate densities directly associated with the longitudinal filaments (e.g. white arrowheads in *Figure 2F*). The punctate densities laterally connect with each other via thin protrusions (yellow arrowheads in *Figure 2F*). This suggests the D-bands form lateral connections that bundle the rootlet filaments.

In the example shown, the rootlet splays into two separate sub-fibers with the space in between them indicated by a black arrowhead (*Figure 2E*). This is consistent with observations of rootlets splaying and merging in the cell (*Spira and Milman, 1979*). In this case, the D1-, D2-, and A-bands are aligned between the two sub-fibers (*Figure 2E*). However, we also observed rootlets where neighboring sub-fibers are offset in their banding pattern. In other words, their D1-bands line up with and connect to neighboring D2-bands (*Figure 2G*, *Figure 2—figure supplement 2A*). A more complicated example of merging sub-fibers is shown in *Figure 2—figure supplement 2B–D*. Here, two sub-fibers (sf1 and sf2) merge with their D1-bands aligned. Sub-fiber 3 (sf3), which lies behind sf1, aligns its D2-band with the D1-band of sf1. In contrast, the fourth sub-fiber (sf4) does not align any D-bands with its neighboring sf2 (*Figure 2—figure supplement 2C, D*).

From a set of 48 tomograms that show sub-fibers, we found D1 aligns with D1 in 22 cases, D1 aligns with D2 in 17 cases and the remainder of cases show no particular alignment between D-bands (9 cases, *Figure 2—figure supplement 2E*). The observation that D1- and D2-bands can align with each other, and their similar morphology raises the possibility that they have the same composition.

## Amorphous bands contain interior density and accumulations on the rootlet surface

The A-bands were visible in central slices but were fainter and more heterogeneous than the D-bands (*Figure 2E*). We found the A-bands also contain lateral connections running through the interior of the rootlet. However, these were both fuzzier and broader than the D-bands and not always picked up in the segmentation (*Figure 2H, I*). Occasionally, these connections appeared as two parallel lines (*Figure 2I*, green arrowheads), but more often their density was discontinuous.

In our tomograms, we see large amorphous accumulations of variable size on the rootlet surface (*Figure 2I, J*, *Figure 2—figure supplement 2F, H*). These correspond to the tufts observed in negative stain EM (*Figure 2C*). When the internal part of the rootlet is masked out these amorphous densities are clearly heterogeneous, but overall follow a banding pattern with 80 nm spacing (*Figure 2J*, *Figure 2—figure supplement 2G–I*). The average position of these surface bands aligns with that of the interior A-band densities (*Figure 2H–J*, *Figure 2—figure supplement 2J*). Additionally, the surface densities show evidence of connecting to the A-bands (*Figure 2I, J*, *Figure 2—figure supplement 2J*).

We did not identify amorphous densities on the surface of rootlets that were surrounded by membranes (*Figure 1C–F*), which suggests these features accumulate during the purification. Since they are on the surface of the rootlet we suspect they derive from the membrane-associated proteins that aggregate upon membrane removal. Their accumulation around the A-band would imply it is the major membrane interacting site.

Overall our analysis of purified rootlet tomograms shows the banding pattern is more complex than previously observed (*Spira and Milman, 1979*; *Gilliam et al., 2012*) and defines three distinct bands (D1, D2, and A) as separate structures within the 80.1 nm repeat.

## Rootlet filaments form a highly flexible network

Fast Fourier transforms of rootlets show strong peaks along the longitudinal axis, corresponding to the banding patterns mentioned above. In contrast, there are no peaks on the lateral axis (*Figure 2—figure supplement 1F*), suggesting longitudinal filaments are not packed together in a regular array. To understand how they do interact we looked in detail at our tomogram segmentations of purified rootlets.

These show the longitudinal filaments all run in a similar direction but are not strictly parallel with each other (*Figure 3A*). We observed two general filament types. One, wider (*Figure 3A–C*, white arrowheads), typically ~5 nm and the other thinner. The thinner type (*Figure 3A–C*, gray arrowheads) were more flexible, often only faintly visible and frequently missed by automatic segmentation (*Figure 3B, C*, orange dotted lines). We see examples where thick filaments appear to 'melt' locally to form small bulges made of two thinner filaments (*Figure 3A, C*, red arrowheads). In other places thick filaments splay apart into two thin filaments (highlighted in pink, *Figure 3A*), which separate and then both merge with other thin filaments (*Figure 3A*, pink arrowhead). These segmentations highlight the flexibility of the longitudinal filaments.

The longitudinal filaments are likely made up of rootletin (*Yang et al., 2002*; *Yang et al., 2005*; *Gilliam et al., 2012*; *Chen et al., 2015*; *Akiyama et al., 2017*). We used AlphaFold2 to predict the structure of dimeric rootletin fragments and found it was confidently predicted as a coiled coil along most of its length (*Figure 3A*, *Figure 3—figure supplement 1*), including the majority of its N-terminus that was previously suggested to be globular (*Yang et al., 2002*; *Akiyama et al., 2017*; *Figure 3—figure supplement 1*). Stitching the overlapping fragments together predicts that rootletin is ~265 nm long and would be expected to extend through three striated repeats of the rootlet (*Figure 3A*). The width of the rootletin dimer did not exceed ~1.3 nm which suggests multiple copies of rootletin may be needed to form the ~5-nm-thick filaments we frequently observed.

## Filament subtomogram average is consistent with two coiled-coil dimers

To characterize the nature of the individual filaments in our tomograms we performed subtomogram averaging. Based on the strong longitudinal regularity in the rootlet, we assumed that each 80.1 nm repeat contains the same features that can potentially be averaged (*Figure 3D*). The bands in these repeats approximate surfaces that section through the rootlet. Keeping in mind that the D1- and D2-bands may tether the filaments together, we expected a higher order of organization along this band. Since the D2-band is closer to the less regular A-band, we experienced difficulties obtaining consistent averages centered around this region. Thus, we extracted subtomograms centered around each D1-band (*Figure 3D*) using tools developed for extracting objects from membrane surfaces (*Qu et al., 2018*; *Burt et al., 2020*; *Leneva et al., 2021*). We manually defined the D1-bands as surfaces in Dynamo (*Castaño-Díez et al., 2017*) and then approximated the number of filaments per surface area. We extracted 591,453 subtomograms from 14 tomograms, approximately four times as many subtomograms as the expected number of filaments. This initial set was rigorously cleaned to discard particles that did not have a filament in their center or had distorted striations, reducing it to 358,863 particles. Further cross-correlation and cluster cleaning yielded a final set of 180,252 particles.

In the initial subtomogram averages, we saw strong densities for the D-bands and longitudinal filaments (*Figure 3E*). A central longitudinal filament density was well defined, although the density for its neighboring filaments was not continuous and did not extend beyond the alignment mask (*Figure 3E*). We attempted to improve the average by using a narrower mask to include only the nearest neighbors of the central filament, but this only marginally improved features (*Figure 3*). To ask if there were any recurring arrangements of neighboring filaments in the data that could allow us to average a homogeneous subset, we resorted to classification of the original set of 591,453 particles (*Supplementary file 1a*, *Figure 3—figure supplement 2A*). The resulting classes described various filament arrangements with a distance between the filament centers ranging from 6.8 to 9.6 nm (*Figure 3—figure supplement 2A*). However, none of the classes formed a regular grid that extended outside of the classification mask. Our data suggest neighboring longitudinal filaments are flexibly associated consistent with our observations in the tomogram segmentation (*Figure 3A*).

Since the filaments have a variable arrangement, high-resolution features of the individual filaments will be obscured in the alignment of a filament array. Therefore, we focused the alignment on

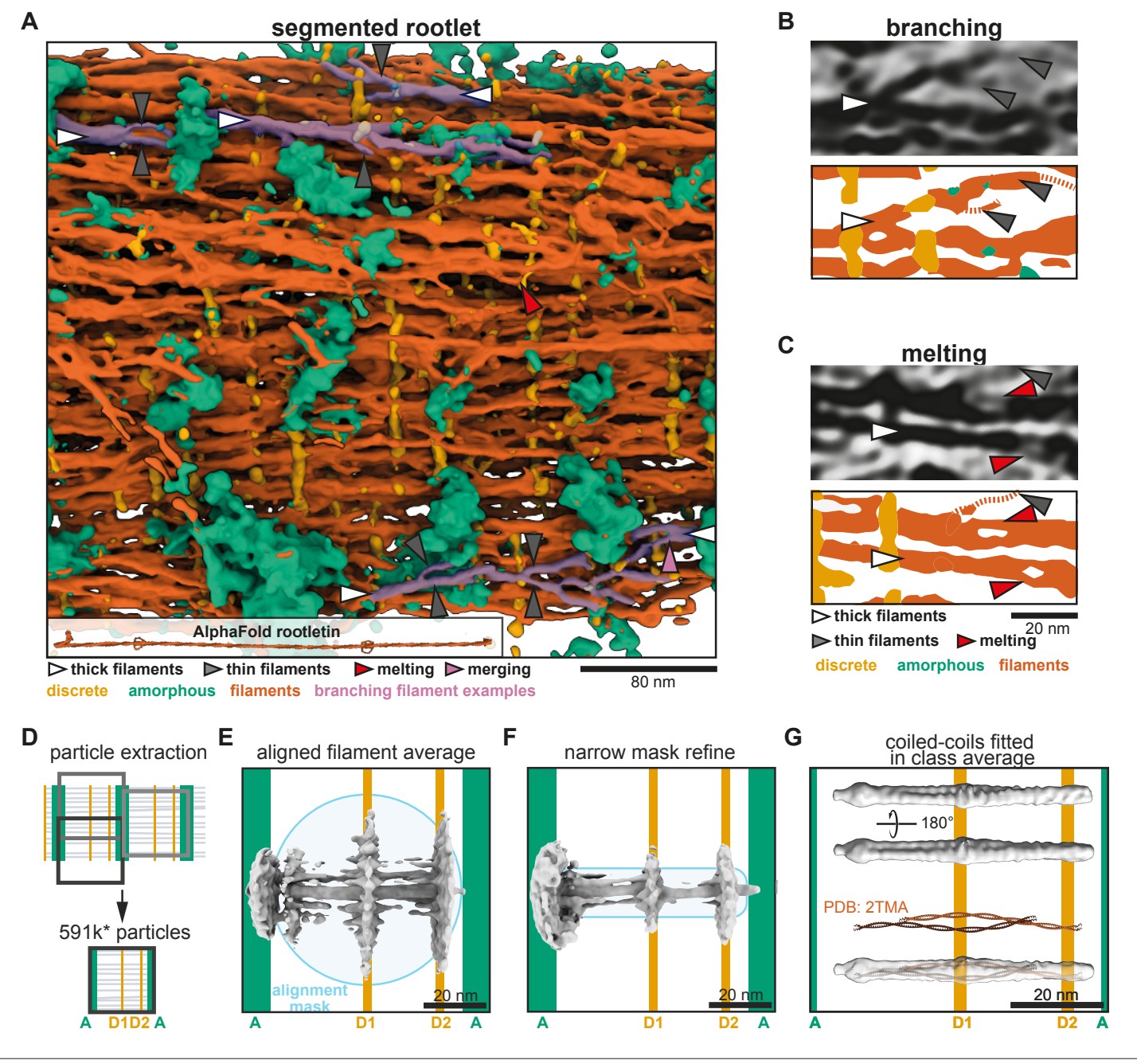

**Figure 3.** Rootlet filaments are highly flexible and occur as coiled-coil dimers. (**A**) Semi-automated segmentation of a rootlet tomogram. The inset shows a 265-nm long model of stitched rootletin AlphaFold predictions. Filaments that show splaying and merging were manually highlighted in pink. (**B, C**) Tomogram slices and their corresponding segmentations. (**B**) Example of thick filaments splaying into thin filaments, each indicated by arrowheads as in the legend of panel C. (**C**) Example of filament melting pointed out by red arrowheads. (**D**) Schematic of the location along the rootlet where particles were extracted. (**E**) The initial average after alignment of 180,252 particles with a wide spherical alignment mask. (**F**) The initial average of particles aligned with a narrower cylindrical mask. (**G**) A class average of 34,490 particles, aligned and classified with a narrow mask. The PDB structure (PDB: 2TMA) of two lamin tetramers is shown in red and fitted in the class average.

The online version of this article includes the following figure supplement(s) for figure 3:

**Figure supplement 1.** AlphaFold predictions of rootletin dimeric fragments.

**Figure supplement 2.** Subtomogram averaging and classification of rootlet filaments.

single filaments to obtain structural information. To be able to average these filaments we avoided using a conventional low-pass filter step in order to preserve the coiled-coil features that are necessary for driving the alignment. We directly unbinned the data to a pixel size of 5.55 Å/pixel and used the rigorously cleaned set of 180,252 particles. To ensure unbiased alignment of any coiled-coil features we generated a smooth reference by randomizing the inplane rotational orientation of the particles (*Figure 3—figure supplement 2B*). Initial refinement of the data resulted in an anisotropic structure since the filaments did not have enough features to align to. Therefore, we performed classification with alignment in RELION 4.0 alpha (*Zivanov et al., 2022*), and used a narrow 3.3-nm-wide mask with a smooth edge up to 7.7 nm (*Figure 3—figure supplement 2B*). This was the narrowest mask that still resulted in an isotropic structure and revealed features that were absent in the smooth reference. The resulting class averages contained a twist along the filament length in classes 2–4 and most prominently in class 5. These four classes contain 72.29% of the particles, highlighting the prevalence of the twist-feature (*Supplementary file 1b*, *Figure 3—figure supplement 2C*). Class 5 contained 19.27% of the data, that is 34,490 particles, and revealed the twist is formed by a filament of 2 nm thick by 5 nm wide with a helical groove along its length (*Figure 3G*).

The groove in the average suggests the filaments consist of two intertwined rods. To see if the density matches the dimensions of multiple coiled-coil dimers, we fitted two copies of the tropomyosin crystal structure into our density (from PDB: 2TMA, *Phillips, 1986*). This was not a perfect fit due to the curvature of the crystal structure, but it showed that our density is consistent with the dimensions of two intertwined coiled coils (*Figure 3G*). Other crystal structures of two coiled-coil dimers (lamin and rootletin CC3, *Figure 3—figure supplement 1D, E*) equally fit well in parts of the density. This suggests the longitudinal filaments contain an oligomerization interface where coiled coils intertwine with each other. This interface extends along the 50 nm length of the average. Taken together, our subtomogram averaging and segmentations suggest that the network of rootlet filaments can be held together by lateral interactions between pairs of coiled-coil dimers.

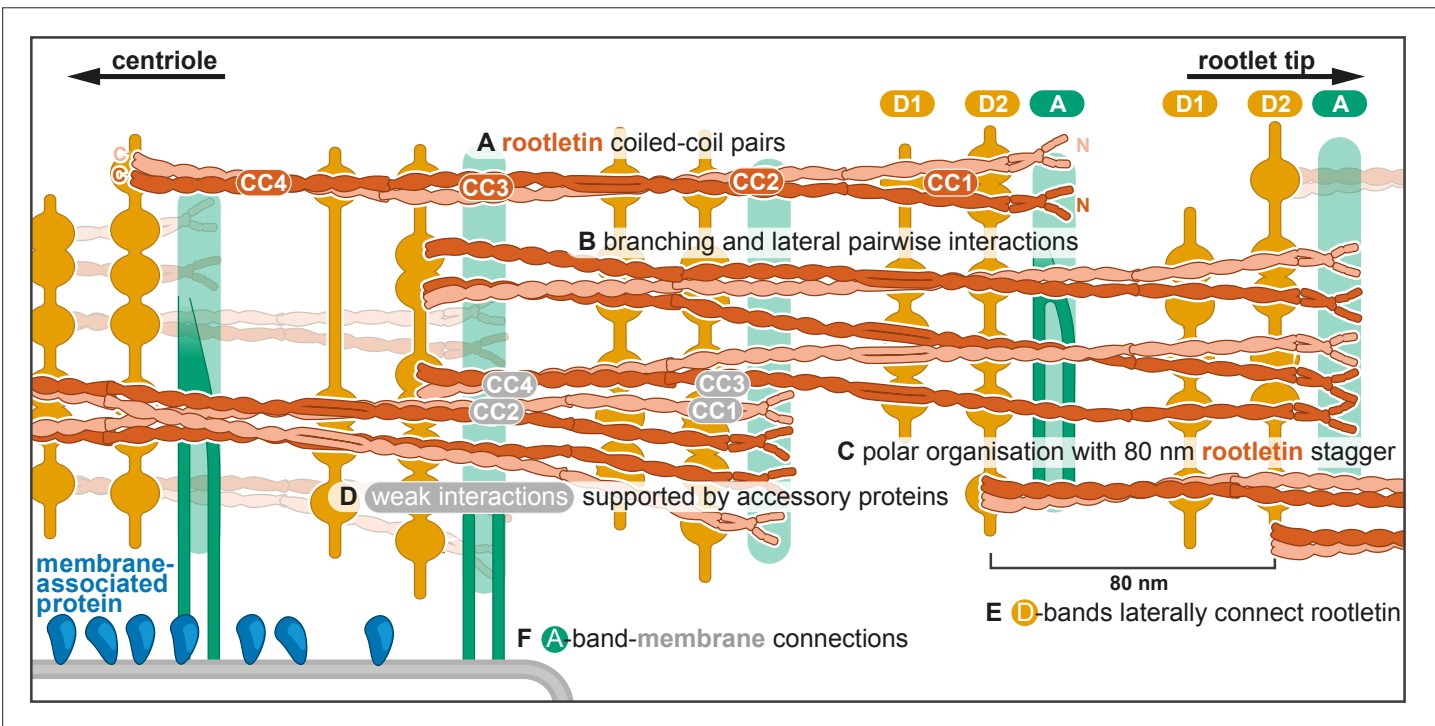

**Figure 4.** Model of rootlet organization. (**A**) Filaments are depicted as rootletin coiled coils based on the 265-nm long AlphaFold prediction. Rootletin coiled coils are shown in different shades of orange for clarity. (**B**) Branchpoints are found where thicker filaments splay into thinner filaments, here indicated as single coiled coils. (**C**) Rootletin molecules are arranged in a polar manner based on the polarity we observed in the striations. The N-termini point away from the centriole, with an offset of 80 nm. We propose the N-termini align with the A-bands. (**D**) Staggered rootletin filaments may be supported via previously reported weak interactions supported by accessory proteins in the D-bands (**E**) D-bands were observed as punctate laterally connected densities associated with filaments. (**F**) Amorphous densities of the A-bands were occasionally observed to contain two parallel lines. A-band accumulations in purified rootlet suggest they correspond to the membrane interaction sites in cellular tomograms.

## Discussion

The metazoan rootlet is a cilium-associated fiber that is characterized by regular cross-striations. The work here allows us to propose a model for its ultrastructure (*Figure 4*) which we will discuss below.

As described above, rootletin has been identified as the main constituent of the rootlet. It consists of four coiled-coil domains (CC1–4) (*Yang et al., 2002*; *Ko et al., 2020*). Crystal structures of CC3 fragments showed them to be obligate homodimers, even stabilized by the presence of disulfide bonds (*Ko et al., 2020*). Initial work suggested the N-terminal region (CC1) may be a globular N-terminus that contributes to the cross-striation pattern (*Yang et al., 2002*; *Akiyama et al., 2017*). Our AlphaFold predictions of dimeric rootletin fragments suggest rootletin forms an extended coiled coil for its entire length. Like other recent studies (*Ko et al., 2020*), we suggest the N-terminus forms a coiled coil instead of a globular domain.

We propose that the thick filaments which run throughout the rootlet are parallel bundles of pairs of rootletin coiled coils (*Figure 4A*). We observed density consistent with this in our subtomogram averaging around the D1 striation. Segmentation of tomograms suggested the thick filaments are found throughout the rootlet 80 nm repeat suggesting the lateral interaction of pairs of coiled coils extends beyond the D1 striation. Furthermore, co-immunoprecipitation (*Ko et al., 2020*) and yeast-2-hybrid studies on rootletin suggest each of the CC2, CC3, and CC4 fragments, but not the N-terminal CC1 fragments, can interact with themselves. In our model, we have therefore shown rootletin coiled coils forming paired bundles between CC2 and CC4, but separating in their CC1 regions (*Figure 4A*).

Our segmentations show that thick filaments can separate into thin filaments, which are then able to merge with other thin filaments to reform thin filaments. We propose that this network formation is due to the ability of individual rootletin coil coils to separate and reform lateral, pairwise, interactions with the relevant part of other rootletin coiled coils (*Figure 4B*).

In the crystal structure of the rootletin CC3 fragment, two pairs of coiled coils pack together in an antiparallel arrangement (*Ko et al., 2020*). Although we cannot rule out such an arrangement, in our model we propose that the rootletin coiled coils all run parallel to each other (*Figure 4C*). This underlying polar arrangement of rootletin filaments most easily accounts for the asymmetric pattern of cross-striations we see in each repeat: where the A-band is closer to the D2-band than the neighboring D1-band (*Figure 4C*). We assume the N-termini of rootletin are pointing away from the centriole as immunogold labeling of the rootletin N- and C-termini that showed the N-termini localized further away from the centrioles (*Bahe et al., 2005*). Thus, we expect that a parallel orientation of rootletin filaments is seeded at the centriole and persists throughout the rootlet filament network.

The 80.1 nm striated repeat in the rootlet is much shorter than the predicted length of rootletin (~265 nm). To allow for such a repeat with a much longer molecule, an individual rootletin may have an 80.1 nm longitudinal offset of its domains to the neighboring rootletin molecule, consistent with Immunofluorescence that found a ~76 nm distance between the N-termini along the rootlet length (*Vlijm et al., 2018*). This creates a model of staggered alignment of rootletin molecules (*Figure 4D*). Co-immunoprecipitation studies show that CC2, CC3, and CC4 only weakly interact with the other coiled-coil domains (*Yang et al., 2002*; *Ko et al., 2020*). Therefore, we propose that there are only minor interactions between the offset rootletin coiled coils and instead they require intermediary proteins (*Figure 4D*).

Although cross-striations are a defining feature of rootlets, previous work did not identify the two distinct types. The D-bands we report most closely match the striations reported in a cellular cryo-ET study of intact human photoreceptor cells (*Gilliam et al., 2012*). However, this previous study did not identify a third band. In contrast, our A-band appears similar to those found by resin-section EM (*Spira and Milman, 1979*; *Yang et al., 2002*), due to their less regular and wider appearance.

The D-bands are formed of punctate densities attached to the longitudinal filaments and are laterally interconnected by elongated densities (*Figure 4E*). Due to the similar appearance and alignment of the D1- and D2-bands, we suggest they may have the same composition. The presence of two D-bands per repeat, in contrast to the rootletin N-terminus that appears once per repeat (*Vlijm et al., 2018*), suggests the D-bands are not composed of rootletin. Our combined findings lead us to propose that the D-bands are formed by a protein, other than rootletin, that bundles rootletin into oligomers (*Figure 4E*).

Candidates for the D-bands are the CEP68 and CCDC102B, which were identified as critical components to assemble and bundle rootlets (*Fang et al., 2014*; *Vlijm et al., 2018*; *Xia et al., 2018*).

The depletion of CCDC102B and CEP68 independently resulted in splayed fibers (*Vlijm et al., 2018*; *Xia et al., 2018*), and a lack of interaction between the two (*Xia et al., 2018*) suggests they form separate rootletin bundling connections. AlphaFold predictions of CEP68 show its C-terminal alpha helices form a three-helix spectrin repeat, while the majority of the protein (residues 1–615) is predicted to be unstructured. Centlein was found to be required for the recruitment of CEP68 to the rootlet (*Fang et al., 2014*). However, this may be through an indirect interaction as our AlphaFold predictions do not find any interaction between CEP68 and any combination of centlein and rootletin fragments.

CCDC102B is a coiled-coil protein that, as a dimer, forms two confidently predicted coiled coils interspaced by a disordered region. It interacts with three different fragments of rootletin in co-immunoprecipitation experiments (*Xia et al., 2018*), indicating a minimum of three separate interaction sites. The extended morphology of CCDC102B is incompatible with the punctate features we observe in the tomograms but would be consistent with the lateral connections between the D-band puncta. Thus, while the identity of the D-band densities remains unknown, we show that they laterally connect twice per 80.1 nm repeat to form a link between interspaced longitudinal filaments (*Figure 4E*).

In our model, we propose that the rootletin N-terminus aligns with the A-band and contains putative membrane connection sites (*Figure 4F*). In striated fibers that formed upon rootletin overexpression, deletion of this N-terminal region removed the single prominent band per striation repeat (*Akiyama et al., 2017*). On the rootlet surface, centered around the A-bands, we found amorphous densities, that were absent in the cellular rootlet tomograms. We propose these amorphous densities accumulated during the purification or are aggregates of the membrane-associated proteins onto major membrane interaction sites in the A-band. Furthermore, in some tomograms, we saw evidence of thin projections extending from the rootlet A-band to contact these amorphous densities.

A number of membrane interactions with the rootlet have been reported, including the nuclear envelope (*Potter et al., 2017*), mitochondria (*Olsson, 1962*; *Hayes et al., 2021*), and membrane saccules (*Spira and Milman, 1979*) which may be the endoplasmic reticulum (*Gilliam et al., 2012*). Potential candidates for making these links are Kinesin Light Chain 3 (KLC3) (*Yang et al., 2002*) and Nesprin1 (*Potter et al., 2017*) which both interact with the rootletin N-terminus. KLC3 interacts with rootletin in CoIP experiments (*Yang and Li, 2005*) and its associated Kinesin heavy chain, Kif5, colocalized with the rootlet in immunofluorescence (*Yang and Li, 2005*). A Nesprin1 isoform, Nesprin1α, connects the rootlet to the nucleus via its C-terminal KASH domain that interacts with the inner nuclear envelope protein SUN2 (*Potter et al., 2017*). Larger isoforms of Nesprin1 were detected along the length of the rootlet in ependymal and tracheal cell types that expressed both KASH-containing and KASH-less variants (*Potter et al., 2017*).

Thin-sectioning EM of rootlet cross-sections previously showed a high variability of overall rootlet shape. They do not form a coherent round bundle but are often split, flattened, indented, or surrounded by mitochondria and membrane compartments (*Yang et al., 2002*; *Hayes et al., 2021*). Moreover, rootlets can splay apart and rejoin (*Hayes et al., 2021*; *Spira and Milman, 1979*), consistent with our observation of their branching into sub-fibers and heterogeneity in classification. This flexibility may provide rootlet fibers the capacity to extend between and tether organelles without restricting organelle shape.

In conclusion, our use of cryo-ET reveals the rootlet to be built of a highly flexible network of filaments connected by three types of cross-striation bands. Cross-striation bands protrude from the rootlet and connect to membranes. The combination of segmentation and subtomogram averaging has let us propose a new model for the rootlet's ultrastructure and its membrane connections that will provide the basis for understanding its function and how it is assembled.

## Methods
### Retinal dissection

Eight to ten eyes from adult pigmented mice were freshly obtained from the in-house animal facility and kept in ice-cold phosphate-buffered saline. All procedures were performed in accordance with UK Home Office regulations and licensed under the UK Animals (Scientific Procedures) Act of 1986 following local ethical approval. The corneas were pierced with sharp tweezers and removed by careful tearing along the corneoscleral junction, followed by the removal of the iris in a similar fashion. The

lens was pulled out of the eye cup, often pulling along a clean retina. Alternatively, the eye cups were crudely pulled apart on an area where the retinas had detached, further exposing the clean retinas.

### Rootlet purification

The rootlet isolation protocol was adapted and optimized based on earlier mouse photoreceptor cell and rootlet purification protocols (*Liu et al., 2007*; *Gilliam et al., 2012*). The retinas were collected in 200 µl Iso-osmotic Ringer's buffer (10 mM HEPES pH 7.4, 130 mM $NaCl_2$, 3.6 mM $KCl_2$, 12 mM $MgCl_2$, 1.2 mM $CaCl_2$, 0.02 mM EDTA) supplemented with 8% Optiprep. The retinas were vortexed for 1 min at 2000 rpm and spun down at $200 \times g$ for 1 min. This process was repeated five times resulting in five supernatants that were layered onto Ringer's buffer containing 10% optiprep with a 30% optiprep cushion. The gradients were centrifuged for 1 hr at $24,700 \times g$ in a TLS-55 swinging bucket rotor. Photoreceptor outer segments were collected from the 10–30% interface, diluted with Ringer's buffer and pelleted in a TLA-100.3 rotor for 30 min at $26,500 \times g$. The pellet was resuspended in Ringer's buffer and imaged on a Nikon TE2000 microscope equipped with DIC prisms and an sCMOS camera. The sample was inspected to optimize the vortexing speed of the retinas for maximal rootlet yields. This sample was then used for EM sample preparation or for further processing: The pellet was resuspended in buffer B (10 mM PIPES, 50 mM KCl, 5 mM $MgCl_2$, 1 mM DTT, 2 mM PMSF, Roche protease inhibitor) with 1% Triton X-100 and incubated on ice for 1 hr. The lysate was layered onto a discontinuous sucrose gradient, containing three steps of buffer B with 40, 50, and 60% sucrose. The rootlets were collected from the pellet, diluted and pelleted at $23,000 \times g$ in a TLA-100.3. Finally, the rootlets were resuspended in buffer B for further analysis by EM.

### EM sample preparation

Samples were vortexed and applied to carbon-supported glow-discharged grids with a thin carbon film (Agar scientific), followed by staining with 2% uranyl acetate. Negatively stained grids were imaged using an FEI 200KeV Tecnai FEG TEM with a Falcon II direct electron detector or FEI Tecnai T12 G2 Spirit with a Gatan Orius SC200B CCD camera.

### Tomography data collection

For imaging by cryo-EM, samples were mixed 1:5 with four times concentrated BSA coated 10 nm gold fiducials (BBI solutions: BSA10). The sample was incubated on glow-discharged Quantifoil 2/2 Au200 grids and plunge-frozen in liquid ethane using an FEI Vitrobot. Grids were screened using an FEI 200KeV Tecnai FEG TEM with a Falcon II before data collection. A dose-symmetric acquisition scheme (*Hagen et al., 2017*) and the updated Serial-EM tilt-series controller (*Xu and Xu, 2021*) were used for the tomography data collection on a Titan-KRIOS III equipped with an FEG emitter and energy-filtered K3 detector used in counting mode. Tomograms of the detergent-extracted rootlets were collected at 1.39 Å/pixel, while 2.23 Å/pixel tomograms were acquired on the rootlets attached to photoreceptor cell outer segments. The cumulative dose per tomogram is 110–120 electrons/$Å^2$.

### Data preprocessing

The raw movies for outer segment rootlet protrusions were preprocessed using the subTOM package as described in *Leneva et al., 2021*, available at https://github.com/DustinMorado/subTOM/ (*Morado, 2021*). This included dose-weighting and motion correction using the IMOD alignframes package (*Kremer et al., 1996*). Subsequent alignment was done using fiducial alignment, solved for rotations and grouped tilt angles with a fixed magnification in Etomo (IMOD) (*Kremer et al., 1996*). Gold beads were erased and bin4 tomograms were reconstructed using weighted back projection. Alternatively, the raw data were preprocessed in WARP (*Tegunov and Cramer, 2019*). This was followed by the Dynamo tilt series alignment (*Castaño-Díez et al., 2017*) implementation for WARP available at https://github.com/alisterburt/autoalign_dynamo (*Burt, 2021*) and final reconstruction in WARP (*Tegunov and Cramer, 2019*).

### Tomogram segmentation

For tomogram segmentation, tomograms were reconstructed as even/odd frame half tomograms using the above-mentioned WARP pipeline and denoised using Noise2Map (*Tegunov and Cramer, 2018*). The tomograms were then deconvolved and isotropically reconstructed with denoising using

IsoNet (*Liu et al., 2021*) with a cube size of 128 pixels. The tomograms were then preprocessed in EMAN2.2 for training of the TomoSeg CNN (*Chen et al., 2017*). Here, the features (filaments, D-bands, A-bands, gold fiducials, actin, membranes, membrane-associated densities, and ice contaminations) were individually trained for each tomogram. This involved manually tracing a training set of 10–20 positive and 100–150 negative boxed areas per feature. We iteratively expanded and curated the training set until the segmentations were accurate, as recommended in the software manuals. Segmented maps were allowed to compete for the assignment of pixels in the tomograms, cleaned up in Amira (Thermo Fisher Scientific) and converted to object files. The object files and corresponding tomograms were displayed in ChimeraX (*Pettersen et al., 2021*). Assignment of direct and indirect striation–membrane connections was done manually by assessing whether TomoSeg-segmented striations and membranes were connected directly or via membrane-associated densities. The automated segmentation of amorphous striations picked up mostly dense amorphous features. The fainter densities that we observed to laterally connect the amorphous features were manually drawn by dotted lines.

## Subtomogram averaging

For particle picking, the tomograms were deconvolved using the TOM package (*Tegunov and Cramer, 2019*). Dynamo was used for particle extraction using the Dynamo surface model (*Castaño-Díez et al., 2012*; *Castaño-Díez et al., 2017*): Each D2 band was traced in multiple slices per rootlet to define dynamo surfaces. Surface triangulation was set to result in 591,453 extraction coordinates approximately four times the number of expected filaments. The coordinates were extracted as a Dynamo table that was subsequently converted to the motl-format using subTOM scripts, available at https://github.com/DustinMorado/subTOM/; *Morado, 2021*; *Leneva et al., 2021*. Particles were extracted from tomograms reconstructed using novaCTF (*Turoňová et al., 2017*).

An initial reference was obtained by in-plane randomizing and averaging all particles prior to alignments. Initial alignments were performed to center filaments, by using a 10-nm wide cylindrical mask, limited to 4 nm shifts in *X* and *Y* with respect to the reference orientation. A spherical mask with large diameter was used for alignments the D-bands, these alignments were restricted to the reference *Z* direction. Particles with no filament in their center, or particles that originated from regions in the rootlet with distorted striations (at the edge of a grid hole) were discarded, resulting in a particle set of 358,863 particles. Cluster- and careful per-tomogram cross-correlation cleaning were applied to remove particle duplicates, remaining particles with no filaments, and particles with disordered D-bands. This resulted in a final cleaned particle dataset of 180,252 particles.

Prior to classification in subTOM, alignments with limited *X/Y/Z* shifts and increasingly finer in-plane rotations were performed on the original dataset with 591,453 particles. Twenty eigenvolumes were generated by *K*-means classification over 20 eigenvectors. The eigenvolumes and particles clustered per eigenvector were assessed to identify which vectors described the missing wedge or structural features (*Leneva et al., 2021*). The structural eigenvectors were used to cluster particles into the final class averages that described particle heterogeneity.

For the final subtomogram class-average that contained the twist, the cleaned particle dataset motl with 180,252 particles was converted to a STAR file compatible with RELION 4.0 alpha (*Zivanov et al., 2022*). Gold beads were removed from the preprocessed tomogram frames by converting the aligned tomogram gold coordinates initially obtained by Etomo bead-finder during preprocessing steps (*Kremer et al., 1996*). Particles were then extracted in RELION 4.0 alpha. The initial reference was an inplane randomized average of the cleaned particle dataset. Instead of refinement, which resulted in anisotropic structures due to a lack of features for the alignment, we used simultaneous alignment and classification. We restricted the alignments to full in-plane rotations with respect to the reference *Z*-axis.

## Striation distance measurements

Tomograms were *Z*-projected through the rootlet and the *Z*-projections were cropped to contain only the rootlet and its cross-striations. The images were Fourier filtered in ImageJ and the striation pattern was thresholded. Next, line profiles were fitted to the longitudinal axis of the rootlet and the intensity profiles were exported to csv files. Fitting of the sinusoids, reporting the sinusoid-fit offset for different bands and generation of the graphs was done using a python script.

## Data visualization

Fourier transforms and the integral image filter: local contrast normalization was performed in ImageJ. AlphaFold predictions of 300 AA long dimer fragments with 50 AA overlap were generated using colabfold 4 that uses a modified version of alphaFold2. To run the large number of sequences we used a customized script called alphascreen (version 1.15) available at https://github.com/sami-chaaban/alphascreen (*Chaaban, 2024*). This setup was used to predict interactions for dimeric and oligomeric combinations of rootletin fragments (e.g. CC2 + CC2, CC3 + CC4, CC1 + CC1 + CC1 + CC1, CC3 + CC3 + CC4 + CC4, etc.). Homodimeric and oligomeric combinations were tested with other proteins identified as putative rootletin binding: CCDC102B, CEP68, beta-catenin, ARL2, and centlein. In our hands, only homodimeric rootletin fragment combinations resulted in confident predictions. The structures were stitched together using UCSF Chimera matchmaker (*Pettersen et al., 2004*). PDB structures were fit into the subtomogram averages and all 3D models were visualized in ChimeraX (*Pettersen et al., 2021*). Cartoons and figures were made using Adobe Illustrator. Contrast of images was adjusted in Adobe Photoshop for good visibility on printed paper.

## Acknowledgements

We thank D Morado for training and support in cryo-ET and subtomogram averaging and S Scheres for early access to Relion 4.0 alpha. We acknowledge K Qu, H Foster, and S Lacey for technical advice. We thank G Manigrasso for feedback on the manuscript; C Ventura, F Abid Ali, S Chaaban, and members of the Carter laboratory for discussion and T Dendooven and A Burt for support with subtomogram averaging.

We thank the MRC Laboratory of Molecular Biology Electron Microscopy Facility for access and support of electron microscopy sample preparation and data collection; J Grimmett and T Darling for providing scientific computing resources; The Mouse Facility for providing mouse eyes. This work was supported by Wellcome [210711/Z/18/Z], the Medical Research Council, as part of United Kingdom Research and Innovation (also known as UK Research and Innovation) [MRC file reference number MC_UP_A025_1011], and C van Hoorn was funded by a Gates Cambridge Scholarship.

## Additional information

### Funding

| Funder | Grant reference number | Author |
| --- | --- | --- |
| Gates Cambridge Trust | | Chris van Hoorn |
| Wellcome Trust | 10.35802/210711 | Andrew P Carter |
| Medical Research Council | MC_UP_A025_1011 | Andrew P Carter |

The funders had no role in study design, data collection, and interpretation, or the decision to submit the work for publication. For the purpose of Open Access, the authors have applied a CC BY public copyright license to any Author Accepted Manuscript version arising from this submission.

### Author contributions

Chris van Hoorn, Conceptualization, Data curation, Software, Formal analysis, Funding acquisition, Validation, Investigation, Visualization, Methodology, Writing - original draft, Project administration, Writing - review and editing; Andrew P Carter, Conceptualization, Resources, Supervision, Funding acquisition, Project administration, Writing - review and editing

### Author ORCIDs

Chris van Hoorn ⓘ https://orcid.org/0000-0002-1319-847X
Andrew P Carter ⓘ https://orcid.org/0000-0001-7292-5430

### Ethics

All procedures were performed in accordance with UK Home Office regulations and licensed under the UK Animals (Scientific Procedures) Act of 1986 following local ethical approval.

Joint Public Review: https://doi.org/10.7554/eLife.91642.4.sa1
Author response https://doi.org/10.7554/eLife.91642.4.sa2

## Additional files

### Supplementary files
• Supplementary file 1. Tables that summarize the particle distribution of two classifications. (a) Particle distribution of subTOM classification for particle heterogeneity. (b) Particle distribution of RELION 4.0 Alpha classification with alignment.

• MDAR checklist

### Data availability
All raw data and filtered bin4 tomograms have been deposited to EMPIAR (EMPIAR-11828), linked to the EMDB accession codes mentioned below. Representative tomograms have been deposited to the EMDB. EMD-18121 is a cellular tomogram used for the segmentation. Tomograms of purified rootlets used for segmentations, striation analysis, and filament analysis are found under EMDB accessions EMD-18122-18125. Segmentation object files, the final class average, striation analysis files, and AlphaFold predictions have been deposited at Biostudies (S-BSST1164, *Sarkans et al., 2018*).

The following datasets were generated:

The following dataset was generated:

| Author(s) | Year | Dataset title | Dataset URL | Database and Identifier |
|---|---|---|---|---|
| van Hoorn C, Carter AP | 2023 | Purified Photoreceptor Cell Rootlet Tomograms | https://doi.org/10.6019/EMPIAR-11828 | EMPIAR, 10.6019/EMPIAR-11828 |
| van Hoorn C, Carter AP | 2023 | A cryo-ET study of ciliary rootlet organization - Example cell-like tomogram | https://www.ebi.ac.uk/emdb/EMD-18121 | Electron Microscopy Data Bank, EMD-18121 |
| van Hoorn C, Carter AP | 2023 | A cryo-ET study of ciliary rootlet organization - purified rootlet example 1 | https://www.ebi.ac.uk/emdb/EMD-18122 | Electron Microscopy Data Bank, EMD-18122 |
| van Hoorn C, Carter AP | 2023 | A cryo-ET study of ciliary rootlet organization - purified rootlet example 2 | https://www.ebi.ac.uk/emdb/EMD-18123 | Electron Microscopy Data Bank, EMD-18123 |
| van Hoorn C, Carter AP | 2023 | A cryo-ET study of ciliary rootlet organization - purified rootlet example 3 | https://www.ebi.ac.uk/emdb/EMD-18124 | Electron Microscopy Data Bank, EMD-18124 |
| van Hoorn C, Carter AP | 2023 | A cryo-ET study of ciliary rootlet organization - purified rootlet example 4 | https://www.ebi.ac.uk/emdb/EMD-18125 | Electron Microscopy Data Bank, EMD-18125 |
| van Hoorn C | 2023 | A cryo-ET study of ciliary rootlet organization | https://www.ebi.ac.uk/biostudies/bioimages/studies/S-BSST1164 | BioImage Archive, S-BSST1164 |

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
