## [Editor Report · eLife assessment]

The study offers a **compelling** molecular model for the organization of rootlets, a critical organelle that links cilia to the basal body, ensuring proper anchoring. While previous research has explored rootlet structure and organization, this study delivers an unprecedented level of resolution, **valuable** to the centrosome and cilia field. This research marks a significant step forward in our understanding of rootlets' molecular organization.

---

## [Referee Report · Joint Public Review]

The study offers a compelling molecular model for the organization of rootlets, a critical organelle that links cilia to the basal body. Striations have been observed in rootlets, but their assembly, composition, and function remain unknown. While previous research has explored rootlet structure and organization, this study delivers an unprecedented level of resolution, valuable to the centrosome and cilia field. The authors isolated rootlets from mice's eyes. They apply EM to partially purified rootlets (first negative stain, then cryoET). From these micrographs, they observed striations along the membranes along the rootlet but no regular spacing was observed.

The thickness of the sample and membranes prevented good contrast in the tomograms. Thus they further purified the rootlets using detergent, which allowed them to obtain cryoET micrographs of the rootlets with greater details. The tomograms were segmented and further processed to improve the features of the rootlet structures. They proposed that a number of proteins, including rootletin, form parallel coiled coils that run along the rootlet longitudinally. They described how the cross-striations form 3 types of periodic structures -D1/D2/A bands- connected perpendicularly to filaments along the length of the rootlets and to membranes. Overall their data provide a detailed model for the molecular organization of the rootlet.

The major strength is that this high-quality study uses state-of-the-art cryo-electron tomography, sub-tomogram averaging, and image analysis to provide a model of the molecular organization of rootlets. The micrographs are exceptional, with excellent contrast and details, which also implies the sample preparation was well optimized to provide excellent samples for cryo-ET. The manuscript is also clear and accessible.

This research marks a significant step forward in our understanding of rootlets' molecular organization.

---

## [Author Response]

The following is the authors’ response to the previous reviews.

**Reviewer #1 (Recommendations for the authors):**
In the revision the authors addressed all the points from this reviewer and most from other reviewers. The method is now described practically and in detail. The only thing this reviewer still misses is number of subtomograms for each structure. How many subtomograms did the authors extract by Dynamo from how many rootlets? How many out of them were valid in K-mean classification and used for sub-averages? Was the subaverage used for training by TomoSeg or each subtomograms belonging to the class? By clarifying it, this work will be referred by those who would take the same approach for other biological structures.

We now added the particle numbers of all structures to the corresponding text, figure legends and methods and elaborate on this below. We also clarify how we trained the TomoSeg network.

Particle numbers:

We extracted 591,453 subtomograms from 14 tomograms. This initial set was rigorously cleaned with Zcleaning, reducing it to 358,863 particles. Further cross-correlation and cluster cleaning yielded a final set of 180,252 particles.

This refined set was used for the structures presented in Figures 3E, F and S5A, B, as well as for the classification shown in Figure S5C. Of the classified particles, 34,490 particles contributed to classaverage 5 in Figure 3G and S5D, E. The detailed particle distribution of this classification is added as a supplementary table:

We further clarified the numbers in the results, method, and supplementary material section:

Results:

Page 7: “Figure 3. … (E) The initial average after alignment of 180,252 particles with a wide spherical alignment mask. (F) The initial average of particles aligned with a narrower cylindrical mask. (G) A class average of 34,490 particles, aligned and classified with a narrow mask.”

Page 7/8: “We manually defined the D1-bands as surfaces in Dynamo (Castaño-Díez et al, 2017) and then approximated the number of filaments per surface area. We extracted 591,453 subtomograms from 14 tomograms, approximately four times as many subtomograms as the expected number of filaments. This initial set was rigorously cleaned to discard particles that did not have a filament in their center or had distorted striations, reducing it to 358,863 particles. Further cross-correlation and cluster cleaning yielded a final set of 180,252 particles.”

Page 8: “We directly unbinned the data to a pixel size of 5.55 Å/pixel and used the rigorously cleaned set of 180,252 particles.”

Page 8: “The resulting class averages contained a twist along the filament length in classes 2, 3 and 4 and most prominently in class 5. These four classes contain 72.29% of the particles, highlighting the prevalence of the twist-feature (Fig S5C, Table S2). Class 5 contained 19.27% of the data, i.e. 34,490 particles, and revealed the twist is formed by a filament of 2 nm thick by 5 nm wide with a helical groove along its length (Fig 3G).”

Methods:

Page 13: “Surface triangulation was set to result in 591,453 extraction coordinates approximately 4 times the number of expected filaments.”

Page 13: “Particles with no filament in their center, or particles that originated from regions in the rootlet with distorted striations (at the edge of a grid hole) were discarded, resulting in a particle set of 358,863 particles. Cluster- and careful per-tomogram cross-correlation cleaning were applied to remove particle duplicates, remaining particles with no filaments, and particles with disordered D-bands. This resulted in a final cleaned particle dataset of 180,252 particles.”

Page 13: “For the final subtomogram class-average that contained the twist, the cleaned particle dataset motl with 180,252 particles was converted to a STAR file compatible with RELION 4.0 Alpha (Zivanov et al, 2022).”

Supplementary material:

Page 17: “Table S1. Particle distribution of RELION 4.0 Alpha classification with alignment.”

Page 22: “Figure S5: (C) Class averages of a classification with alignment of particles from Fig S5A. Their particle distribution is shown in Table S2.”

For the initial classification, to identify a homogeneous subset, we used the original set of 591,453 picked particles (Fig S5A). The class distribution for this set is added as a supplementary table.

We further clarified this in the results, methods and supplementary material:

Results:

Page 8: “To ask if there were any recurring arrangements of neighboring filaments in the data that could allow us to average a homogeneous subset, we resorted to classification of the original set of 591,453 particles (Fig S5A, Table S1).”

Methods:

Page 13: “Prior to classification in subTOM, alignments with limited X/Y/Z shifts and increasingly finer in-plane rotations were performed on the original dataset with 591,453 particles.”

Supplementary material:

Page 17: “Table S2. Particle distribution of subTOM classification for particle heterogeneity.”

Page 22: “Figure S5: … The surfaces of a cross-section through the filament classes are shown in orange. The particle distribution is provided in Table S1. (B) …”

TomoSeg network training

The subtomograms and the class averages presented at the end of the manuscript were not used as input for training the TomoSeg network. TomoSeg training requires positive and negative sets of segmented 2D regions of interest within tomogram slices. These areas were selected and segmented within the Eman2 TomoSeg GUI, iteratively increasing the size of the training sets until satisfactory performance was achieved.

We have clarified the TomoSeg training process in the methods section to avoid confusion:

Methods:

Page 13: “The tomograms were then preprocessed in EMAN2.2 for training of the TomoSeg CNN (Chen et al, 2017). Here, the features (filaments, D-bands, A-bands, gold fiducials, actin, membranes, membrane-associated densities and ice contaminations) were individually trained for each tomogram. This involved manually tracing a training set of 10-20 positive and 100-150 negative boxed areas per feature. We iteratively expanded and curated the training set until the segmentations were accurate, as recommended in the software manuals. Segmented maps were allowed to compete for the assignment of pixels in the tomograms, cleaned up in Amira (Thermo Fisher Scientific) and converted to object files.”